# Heterogeneity of malaria transmission in urban settings in Ethiopia: A seroprevalence and risk factor analysis

Hiwot Teka[1,2]*, Saron Fekadu[2,3], Legesse Alamerie Ejigu[2], Solomon Sisay[2], Melat Abdo[2], Mulugeta Demisse[2], Hassen Mamo[3], Chris Drakeley[4], Isabel Byrne[4], Girmay Medhin[5], Lemu Golassa[5], Endalamaw Gadisa[2], Fitsum Girma Tadesse[2]

1 Department of Microbiology, Immunology and Parasitology, Addis Ababa University, Addis Ababa, Ethiopia, 2 Malaria and Neglected Tropical Diseases Department, Armauer Hansen Research Institute, Addis Ababa, Ethiopia, 3 Department of Microbial Sciences and Genetics, Addis Ababa University, Addis Ababa, Ethiopia, 4 Department of Infection Biology, London School of Hygiene and Tropical Medicine, London, United Kingdom, 5 Medical Parasitology Department, Aklilu Lemma Institute of Health Research, Addis Ababa University, Addis Ababa, Ethiopia

* hiwotteka2@gmail.com

## Abstract

Malaria remains a significant public health concern in Ethiopia, affecting both rural and urban populations. In urban settings, rapid unplanned expansion, poor drainage, and population movement are increasingly creating aquatic habitats that support vectors such as *Anopheles stephensi,* thereby leading to complex and underestimated transmission patterns. Serology-based risk factor analyses offer a sensitive indicator of recent and past exposure, helping to reveal underlying transmission patterns and more tailored interventions. This study assessed malaria seroprevalence and associated risk factors among residents of three towns in Ethiopia: Adama, Metehara, and Awash Sebat Kilo to better characterize transmission dynamics and identify context-specific vulnerabilities. A community-based cross-sectional survey was conducted from January 30 to March 14, 2020, involving 912 individuals from randomly selected 196 households. Malaria infection was diagnosed using rapid diagnostic tests (RDTs) and quantitative polymerase-chain reaction (qPCR). Serological analyses were performed to determine exposure to *Plasmodium falciparum* and *Plasmodium vivax*. The study found a low malaria prevalence of 0.78% by RDT and 4% by qPCR, with 1.7% of cases due to *P. falciparum* and 2.3% due to *P. vivax*. The overall seroprevalence was 19.4% (149/767) for *P. falciparum* and 18% (139/767) for *P. vivax* using combined short-lived response markers. Seroprevalence rates varied among towns, with Awash Sebat Kilo showing the highest exposure for recent markers (36.5% for *P. falciparum*; 28% for *P. vivax*) compared to lower rates in Adama (2.8% and 2.8%, respectively). Proximity to breeding sites (OR = 2.8, 95% CI: 1.1–7.4; p = 0.037) and being older than 15 years (OR = 4.0 95% CI: 2.1–7.6; p < 0.001) were significant risk factors for *P. falciparum* and *P. vivax* malaria exposure,

**Data availability statement:** All the data used in this work will be available on Dryad (Dryad Digital Repository. DOI: 10.5061/dryad.pc866t227). The R codes used to run the analyses reported in this study available at https://github.com/AHRI-Malaria/Urban_malaria_serology.

**Funding:** NOARD and Sida core funding for Armauer Hansen Research Institute. The funders had no role in study design, data collection and analysis, decision to publish, or preparation of the manuscript.

**Competing interests:** The authors declared no competing interest.

respectively. This study demonstrates marked heterogeneity in malaria exposure across urban settings, with the lowest seroprevalence observed in Adama and higher levels in Awash Sebat Kilo. The findings emphasize the importance of incorporating serological surveillance into routine monitoring and highlight the need for targeted, locally tailored control strategies. Such tailored approaches will be increasingly critical, particularly in the context of emerging urban transmission risks driven by the expansion of *An. stephensi* in the region.

## Introduction

Despite notable progress in malaria control since the early 2000s, the disease remains a major global health challenge. In 2023, an estimated 263 million cases and 597,000 deaths were reported globally, representing an increase of 11 million cases compared to the previous year [1]. The African region continues to bear a disproportionate burden, with Ethiopia ranking among the top five countries in terms of malaria cases, contributing approximately 4% of the global caseload [1].

A complex interplay of ecological and socio-environmental factors shapes malaria transmission in Ethiopia. Seasonal rainfall and temperature variations play a central role in shaping transmission dynamics [2]. Additionally, human-driven changes such as mass settlement, irrigation [3,4], and population mobility [5] further influence transmission intensity and spatial distribution.

The country is home for diverse malaria vector species, *Anopheles arabiensis*, *An. funestus*, *An. pharoensis*, *An. nili, An. coustani* and the recently detected *An. stephensi* [6–9], an urban-adapted invasive vector with significant public health implications. Before 2019, Ethiopia achieved a reduction of malaria cases and death by greater than 50% compared to its baseline in 2015; however, recent years have seen a reversal of these gains due to biological threats and disruption of control efforts caused by ongoing conflict [10,11].

According to Ethiopia's Health Management Information System (HMIS), 5,677,912 cases of malaria were reported, accounting for 13% of all nationally reported disease cases in 2023 [12]. Of the confirmed cases, *Plasmodium falciparum* accounted for 67%, while *Plasmodium vivax* contributed 33%. Children under five accounted for 15.3% of confirmed cases, and 2.1% of cases required hospitalization [13].

Although malaria is considered a rural disease, it is increasingly affecting urban populations. According to the 2015 Malaria Indicator Survey conducted in Ethiopia, malaria prevalence was estimated at 0.6% in urban areas compared to 1.2% in rural settings [14]. Surveillance data from 2014 to 2019 showed that approximately 15% of nationally reported malaria cases were from urban districts [15]. With Ethiopia's population projected to reach 190 million by 2050, and 40% of this population expected to reside in urban areas, thus urban malaria burden will rise substantially [16]. Contributing factors include rapid unplanned urbanization [17,18], increased rural-to-urban population migration [19], and the emergence and expansion of *An. stephensi,* which thrives in urban habitats [20].

Despite the recent setback in malaria control, Ethiopia remains committed to achieving malaria elimination by 2030 [21]. One of the pivotal strategies for achieving this target is the timely detection and treatment of all malaria infections through a better understanding of their spatial and temporal distributions. Understanding malaria risk factors in urban residents will inform targeting of interventions and improve impact [22]. This study aims to assess the sero-prevalence of malaria and identify associated risk factors among urban residents in three towns in Ethiopia: Adama, Metehara and Awash Sebat Kilo.

## Materials and methods

### Study area

This study was conducted in three Ethiopian towns situated within the Rift Valley in Adama, Metehara, and Awash Sebat Kilo, located along the main transportation route from Addis Ababa to Djibouti port (Fig 1).

Adama, situated in central Ethiopia (10°42'0"N, 39°34'0"E) within the Oromia Regional State, lies 92 km southeast of Addis Ababa at an elevation of 1,712 meters above sea level. In 2017, it had an estimated population of 355,475 and a

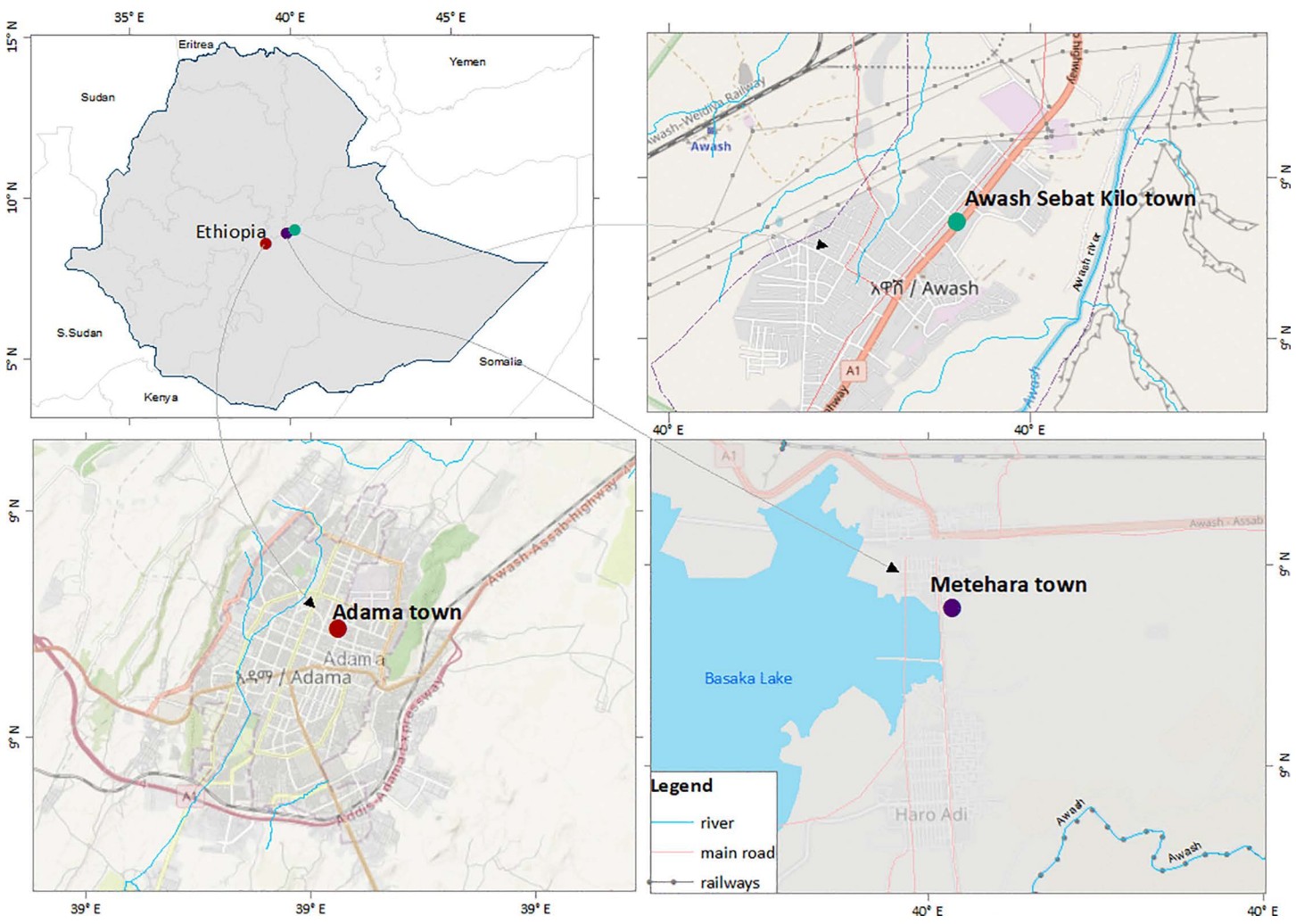

**Fig 1.  Map of the study sites [23].**

population density of 2,660/km$^2$ over its 133.6 km$^2$ area [24]. The town has a semi-arid climate, receiving precipitation of 809 mm annually, with an average temperature of 20.5˚C. The city of Adama serves as a significant transit and industrial hub. There are three major sugar corporations with substantial irrigation activities in the surrounding areas. Moreover, investments in wind energy and industrial parks within Adama have attracted a considerable labour force, estimated at 80,000 individuals [25,26].

Metehara is located at 131 km from Addis Ababa at 08°54′N, 39°55′E and serves as an administrative centre for the Fentale District, East Shoa Zone of Oromia. Positioned at 947 meters above sea level, the town has a semi-arid climate with 847 mm average annual rainfall and a mean temperature of 25.5°C. With an estimated population of 110,961 in 2017 [24], Metehara is home to the country's third-largest sugar factory, which cultivates over 10,000 hectares of sugarcane using irrigation from the Awash River.

Awash Sebat Kilo town situated 31 km from Metehara at 8°58′59.99"N, 40°10′0.01"E and located in Zone 3 of Afar Region. It has an altitude of 916 meters above sea level. It has a semi-arid climate with an estimated 567 mm annual average rainfall and an average temperature of 25.8°C. The Awash River is the primary source of irrigation. In 2017, the town's population was estimated at 46,909 people [24].

Malaria remains a major public health concern in these towns. Routine surveillance data was obtained on November 11, 2019 from Ethiopian Public Health Institute. In 2019 the surveillance data showed 22,856 (Adama), 21,622 (Metehara), and 4,448 (Awash Sebat Kilo) febrile patients examined at health facilities of which 1,056 (4.6%), 3,695(17.1%), and 501(11.3%) tested positive for malaria, respectively. *Plasmodium falciparum* was the predominant species, accounting for 57% of infections in Adama, 74% in Metehara, and 76% in Awash Sebat Kilo.

A time series analysis (2014–2019) showed stable malaria incidence in Metehara and Awash Sebat Kilo, while Adama exhibited a significant decline [15] (see S1 Fig). Based on World Health Organization stratification, all three towns are considered low transmission settings [27]. However, national stratification places Metehara in a high transmission stratum (an annual parasite incidence (API) of 52.8, cases/1,000 population at risk); Awash Sebat Kilo as moderate (API=with 41/1,000); and Adama as very low (API = 2.5/1,000) [15,28].

## Sampling design, sampling technique and study population

Sample size was calculated using the population proportion formula, assuming a 95% confidence interval (95% CI) and a design effect of 1.5. The target was to detect a polymerase chain reaction (PCR)-based malaria prevalence of 10% [29], incorporating a 20% refusal rate and an average household size of 3 [30]. A total of 248 individuals from 83 urban households were estimated to be enrolled from each of the three towns.

A community-based cross-sectional survey was conducted in dry season between January 30 and March 14, 2020 to assess malaria infection and exposure across all age groups. GPS coordinates were generated from a 100-meter gridded map of each town using ArcGIS (version 10.8). Eighty-three coordinates were then selected using simple random sampling in Stata software14 (StataCorp, TX, USA), and subsequently matched to the closest households for the survey. Field teams navigated to selected household coordinate using GPS (Garmin-garmin GPSMAP® 64) device.

All household residents aged over 6 months were eligible to be included in the study. Following a thorough explanation of the study purpose, written informed consent was obtained from adult participants and the guardians of minors. Additionally, assent was obtained from children aged 12–17 years after they were given the opportunity to decide whether they wished to participate. Structured questionnaires preloaded on tablets (Samsung SM-T377a Galaxy) with GPS functionality were used to collect data on demographics, malaria knowledge and perceptions, history of travel to malaria endemic areas, malaria prevention intervention utilization, and care-seeking behaviour.

The household head was interviewed, and all eligible individuals consenting to blood collection were tested using a malaria rapid diagnostic test (RDT) (CareStartTM, Malaria *Pf*/*Pv* (HRP2/pLDH). Additionally, 100 µL of blood obtained from fingerpick into microtainer tubes (BD K$_2$EDTA) was used to make dried blood spots (DBS) on filter paper (903

Whatman, Maidstone, UK) for PCR and serology tests. RDT-positive cases were treated according to national treatment guidelines [31]. DBS were air-dried, stored in with desiccant beads in sealed bags (Geejay Chemicals Ltd), and transported at ambient temperature to the Armauer Hansen Research Institute (AHRI) laboratory in Addis Ababa for storage at −20°C.

## Molecular and serological analyses

Genomic DNA was extracted using DNeasy Blood & Tissue, 250 QIA mini-Kit (QIAGEN, cat. nos. 69504 and 69506) from a 6 mm DBS disc according to the manufacturers protocol [32]. TaqMan-based quantitative PCR using the Fast-Advanced master mix (Applied Biosystems, #4444557) targeting 18S rRNA of *P. falciparum* and *P. vivax* parasites. Species-specific primers and probes were used on BioRad CFX-2000 as described before [33,34]. Parasite gene copy number was estimated using a standard curve generated from serial dilutions of recombinant plasmids. Parasite density was estimated from the gene copy number following quantification models described before [34]. The dilution factor from blood to DNA eluate considered the estimated blood volume eluted within the 6 mm diameter spot area as described by Corran *et al.* [35]. For each experiment, known and quantified patient blood samples were used as negative and positive controls while nuclease-free water was used as non-template control. A sample was considered negative if there was no detectable signal or signal above the cut-off value (*i.e.,* Cycle threshold value of 37.2 for *P. vivax* and 36.4 for *P. falciparum*).

## Serum extraction and bead-based immune assay

Serum was eluted from 6 mm DBS punches using 150 μL PBS containing 1% sodium azide, 0.5% Tween, and 0.5% bovine serum albumin (BSA), yielding a 1:36.6 dilution, after an overnight shaking at room temperature. Eluates were stored at −20°C until analysis. Samples were further diluted 1:400 in Buffer B and stored at 4°C overnight.

Antibody responses to 10 malaria antigens were measured using MAGPIX® as described before [36], covering long- and short-lived exposure markers for both *P. falciparum* and *P. vivax* (see S1 Table). Antigen-coupled beads [37] were incubated with samples and PE-labeled anti-IgG, followed by fluorescence reading. Each plate included negative controls (European naïve serum, local RDT-negative sera). *P. falciparum* positive serum control prepared from a hyper-immune individual (CP3) from Tanzania was included as a positive control in 5-fold serial dilution from 10-fold pre-dilution. The WHO international standard serum for both *P. falciparum* and *P. vivax*, NIBSC 10/198 and NIBSC 19/198, respectively, were used as quality control for the assay. To maintain the stability of the controls, the same batch was used for all assay plates and stored at +4°C to avoid the freeze-thaw cycle.

## Statistical analysis

Descriptive statistics were used to summarize the data on sociodemographic, knowledge, fever, and malaria prevalence. A two-component finite mixture model (FMM) with normal distribution and maximum likelihood estimation outputs for component means and variances was applied to the median fluorescence intensity (MFI) values within each group to facilitate the classification of immune responses to sero positive or negative [38–41]. Combination of *P. falciparum* antigens, PfAMA1, PfMSP1_19, and GLURP R2 [42] as long-lived response markers and Etramp5Ag1 GEXP18 and HSP40. Ag1 as short-lived response markers; [43–45] and for *P. vivax* antigens, Pv DBP RII and PvAMA1 as long-lived response markers, PvMSP119, and PvEBP as short-lived response markers were used [46]. Analysis of seroprevalence was carried out using reversible catalytic models (RCM). These models allow for the estimation of seroconversion rates, which quantify the transmission intensity and correspond to the rate at which individuals convert from seronegative to seropositive through exposure to malaria parasites over time [42,47]. The model was fitted independently for each site (Adama, Metehara, and Awash Sebat Kilo).

Generalized Estimating Equations (GEE) modelling approach was used to assess potential risk factors for malaria infection exposure using the short-lived response markers [48]. To account for the binary nature of the data, the response variable was modelled with a binomial distribution and a logit link function. This method allowed us to look at the relationship between the predictor variables and the likelihood of being seropositive for short-lived response markers for both *P. falciparum* and *P. vivax,* while controlling for potential clustering effects within towns and households. Initially, 14 variables were considered, including age, gender, educational level, travel history, wealth index of the household, at least own one ITN, sleeping under an ITN the previous night, knowledge about malaria symptoms, knowledge about mode of transmission, water sources, housing structure (roof material, flooring material and eave structure), and household distance from permanent breeding site (stream or river within 1 km radius). To establish a reliable variable selection process, a p-value of less than or equal to 0.25 was used for variable inclusion.

To examine the spatial distribution of recent malaria infection exposure, spatial autocorrelation analysis was first conducted using the Global Moran's I statistic [49]. This method assesses whether malaria seropositive for recent markers are randomly distributed or exhibit a significant spatial clustering patterns across households [50]. Subsequently, Anselin Local Moran's I was used via the Cluster and Outlier Analysis tool in ArcGIS (version 10.8) to identify local clusters and spatial outliers. Positive higher z-score indicated a heightened concentration of cases, signifying a hot spot; and conversely negative smaller z-score indicated a heightened concentration of lower cases, indicating a cold spot. All spatial analyses and map outputs were generated using ArcGIS (version 10.8).

Statistical analyses were carried out using STATA software version 17 (StatCorp, TX, USA) and R version 4.3.3 [51].

### Ethical clearance

Ethical approval was obtained from the Aklilu Lema Institute of Pathobiology Institutional Review Board, Addis Ababa University (Ref. No: ALIPB IRB/025/2011/2019) and Armauer Hansen Research Institute (AHRI)/Africa Leprosy Tuberculosis, Rehabilitation and Training Hospital (ALERT) Ethics Review committee (Protocol number: P007/19). Written informed consent from adults and guardians of children, and assent for children between the age of 12–17 were obtained after they received a full explanation of the study's purpose, procedures, risks, and benefits. Prospective data were de-identified to maintain participant confidentiality. The retrospective, surveillance data used for S1 Fig were obtained as aggregated case numbers thus individuals cannot be identified.

## Results

### Household and sociodemographic characteristics of the study participants

Of the 277 households approached, 71% (n = 196) were successfully surveyed. The remaining 81 households were not included due to refusal (n = 13), absence of residents (n = 14), or no dwelling in the proximity of the selected GPS coordinate (n = 54). Among the surveyed households, 12% (n = 23) had agricultural land and 32% (n = 62) owned livestock. Surveyed households had access to improved drinking water 99% (n = 194), electricity 89% (n = 175), and sanitation facilities 81% (n = 158). However, less than half of the households 41% (n = 81) had improved cooking facility. Most households owned mobile phones 95% (n = 186) and a television 83.7% (n = 164). At least one ITN was present in 72% of households (n = 141), with >80% coverage in Metehara and Awash Sebat Killo Towns. Among the households who owned ITNs 52% (n = 47) had them visibly hanging. Indoor residual spraying (IRS) in the prior 12 months was reported in few households 8.2% (n = 16). Most households had improved roofing 87% (n = 171), flooring 76% (n = 149) materials and had closed eaves 68.9% (n = 135). Households were evenly distributed across the wealth quantiles, with 20% (n = 39) residing in the lowest quantile (Table 1).

A total of 912 individuals were included in the surveyed households; of whom 51.2% (n = 467) were male. The median age was 21 years (Inter Quartile range: 9–34), and 16.8% (n = 153) were under the age of 5 years. About 18.6% (n = 170) of participant have no occupation. A third of participants, 36.4% (n = 332) reported no formal education, however 42%

**Table 1. Household characteristics: Distribution of selected household characteristics by city/town.**

| Characteristics | City/Town | | | Total n(%) |
|---|---|---|---|---|
| | Adama n (%) | Metehara n (%) | Awash Sebat Kilo n (%) | |
| **Total number of households (HH)** | 85 | 82 | 110 | 277 |
| Surveyed | 62 (72.9) | 71 (86.5) | 63 (57.3) | 196 (70.9) |
| Not Surveyed | 23 (27.1) | 11 (13.4) | 47 (42.7) | 81 (29.2) |
| **Mean HH members** | 4.8 | 4.3 | 5 | 4.7 |
| **Average number of sleeping spaces** | 2.4 | 2.2 | 2.9 | 2.4 |
| **Own at least one ITNs** | | | | |
| Yes | 27 (43.5) | 57 (80.3) | 57 (90.5) | 141 (71.9) |
| No | 35 (56.5) | 14 (19.7) | 6 (9.5) | 55 (28.1) |
| **IRS (in the past 12 months)** | | | | |
| Yes | 4 (6.5) | 3 (4.2) | 9 (14.3) | 16 (8.2) |
| No | 58 (93.6) | 68 (95.8) | 54 (85.7) | 180 (91) |
| **HH Eaves** | | | | |
| Open eaves | 14 (22.6) | 31 (43.7) | 16 (25.4) | 61 (31.1) |
| Closed eaves | 48 (77.4) | 40 (56.3) | 47 (74.6) | 135 (68.9) |
| **Electricity** | | | | |
| Yes | 47 (75.8) | 67 (94.4) | 61 (96.8) | 175 (89.3) |
| No | 15 (24.2) | 4 (5.6) | 2 (3.2) | 21 (10.7) |
| **Roof material** | | | | |
| Improved | 54 (87.1) | 60 (84.5) | 57 (90.5) | 171 (87.2) |
| Not improved | 8 (12.9) | 11 (15.5) | 6 (9.5) | 25 (12.8) |
| **Floor material** | | | | |
| Improved | 47 (75.8) | 58 (81.7) | 44 (69.8) | 149 (76) |
| Not improved | 15 (24.2) | 13 (18.3) | 19 (30.2) | 47 (24) |
| **Sanitation facility** | | | | |
| Improved | 48 (77.4) | 63 (88.7) | 47 (74.6) | 158 (80.6) |
| Not improved | 14 (22.6) | 8 (11.3) | 16 (25.4) | 38 (19.4) |
| **Cooking fuel** | | | | |
| Improved | 33 (53.2) | 31 (43.7) | 17 (27) | 81 (41.3) |
| Not improved | 29 (46.8) | 40 (56.3) | 46 (73) | 115 (58.7) |
| **Source of drinking water** | | | | |
| Improved | 61 (98.4) | 71 (100) | 62 (98.4) | 194 (99) |
| Not improved | 1 (1.6) | 0 (0) | 1 (1.6) | 2 (1) |
| **Location of drinking water** | | | | |
| Water on premises | 37 (59.7) | 60 (84.5) | 46 (73) | 143 (73) |
| Neighbours dwelling | 18 (29) | 11 (15.5) | 9 (14.3) | 38 (19.4) |
| Other | 7 (11.3) | 0 (0) | 8 (12.7) | 15 (7.7) |
| **Household assets** | | | | |
| Radio | 28 (45.2) | 32 (45.1) | 33 (52.4) | 93 (47.5) |
| Television | 49 (79) | 65 (91.6) | 50 (79.4) | 164 (83.7) |
| Non-mobile phone | 5 (8.1) | 7 (9.9) | 1 (1.6) | 13 (6.6) |
| Mobile phone | 57 (91.9) | 68 (95.8) | 61 (96.8) | 186 (94.9) |
| Computer | 8 (1.9) | 3 (4.2) | 4 (6.4) | 15 (7.7) |
| Refrigerator | 31 (50) | 47 (66.2) | 25 (39.7) | 103 (52.6) |
| Bicycle | 6 (9.7) | 19 (26.8) | 9 (14.3) | 34 (17.4) |
| Motorcycle | 5 (8.1) | 4 (5.6) | 3 (4.8) | 12 (6.1) |

*(Continued)*

**Table 1.** (Continued)

| Characteristics | City/Town | | | Total n(%) |
|---|---|---|---|---|
| | Adama n (%) | Metehara n (%) | Awash Sebat Kilo n (%) | |
| Cart | 5 (8.1) | 6 (8.5) | 2 (3.2) | 13 (6.6) |
| Car or truck | 6 (9.7) | 6 (8.5) | 7 (11.1) | 19 (9.7) |
| Ownership of agricultural land | 19 (30.7) | 2 (2.8) | 2 (3.2) | 23 (11.7) |
| Ownership of livestock | 25 (40.3) | 24 (33.8) | 13 (20.6) | 62 (31.6) |
| **Wealth Quintile** | | | | |
| Lowest | 15 (24.6) | 8 (11.6) | 16 (25.4) | 39 (20.2) |
| Second | 10 (16.4) | 16 (23.2) | 20 (31.8) | 46 (23.8) |
| Third | 10 (16.4) | 13 (18.8) | 8 (12.7) | 31 (16.1) |
| Fourth | 8 (13.1) | 19 (27.5) | 12 (19.5) | 39 (20.2) |
| Highest | 18 (29.5) | 13 (18.8) | 7 (11.1) | 38 (19.7) |

(n = 379) had secondary schooling or higher. Among the total of 736 participants, 78.5% (n = 578) of the participants know about fever as malaria symptom and 89.4% (n = 658) knew its mode of transmission by mosquitoes (Table 2). In the preceding two weeks, 10% (n = 91) of participants reported that they had fever and 19% (n = 176) had travelled to other malarious districts in the previous month. About half of the participants 48.6% (n = 443) reported to have used ITN in the previous night.

## Prevalence of malaria infection

Among 774 individuals tested by RDT, only 0.78% (n = 6) tested positive: 0.65% (n = 5) for *P. falciparum* and 1 (0.13%) for mixed species *P. falciparum* and *P. vivax* infection. One case was in a child aged 5 (Table 3). Quantitative PCR (qPCR) conducted on 773 individuals identified 4% infections (n = 31): 2.3% *P. vivax* (n = 18), and 1.7% *P. falciparum* (n = 13). Mean parasite densities were 12.4 parasites/µL (interquartile range [IQR]: 8.5–22.2) for *P. vivax* and 413 parasites/µL (IQR: 62.1–1826.8) for *P. falciparum* (Table 3). Among qPCR positives, 25.8% (n = 8) were under 5 years old, and 67.7% (n = 21) were male. Notably, 72.2% (n = 13) of the *P. vivax* cases were from Adama.

## Seroprevalence and seroconversion

Serological testing on 767 samples revealed seroprevalence of 19.4% (n = 149) for *P. falciparum* and 18% (n = 139) *P. vivax* using combined short-lived response markers. Awash Sebat Kilo had the highest prevalence for most markers, with combined rates of 36.5% (n = 96) for *P. falciparum* and 28% (n = 74) for *P. vivax*, followed by Metehara. Adama recorded the lowest prevalence—2.8% (n = 7) for both *P. falciparum* and *P. vivax*. Long-lived marker analysis also revealed higher overall seroprevalence: 33.9% (n = 260) for *P. falciparum* and 23.5% (n = 180) for *P. vivax*. The same geographic pattern persisted, with Awash Sebat Kilo again showing the highest rates, followed by Metehara, and Adama remaining the lowest (5.7% for both *P. falciparum* and *P. vivax*) (S2 Table and Table 3).

The estimated seroconversion rates for long-lived *P. falciparum* markers ranged from 0.015 to 0.035 in Adama and Awash Sebat Kilo, while for *P. vivax* rates varied between 0.002 and 0.063 in Adama and Metehara respectively (see Fig 2). To assess the relationship between age and seroprevalence, reverse catalytic model with two forces of infection were employed, suggesting a clear age-dependent increase in seropositivity in Metehara and Awash Sebat Kilo when using the combined long-lived markers for *P. falciparum* (PfAMA1 PfMSP119 and PfGLURP2) (Fig 2a and 2b), suggesting continued exposure over time. In Metehara, the notably low *P. vivax* seropositivity among individuals under 20 years suggests a marked decline in malaria transmission within the past two decades. In contrast, data from Awash Sebat Kilo did not show evidence of recent transmission reduction, pointing

**Table 2. Demographic characteristics of the study participants.**

| Sociodemographic Characteristics | City/Town | | | |
|---|---|---|---|---|
| | **Adama n (%)** | **Awash Sebat Kilo n (%)** | **Metehara n (%)** | **Total n (%)** |
| Number of individuals | **302** | **306** | **304** | **912** |
| **Sex** | | | | |
| Male | 161 (53.3) | 158 (51.6) | 148 (48.7) | 467 (51.2) |
| Female | 141 (46.7) | 148 (48.4) | 156 (51.3) | 445 (48.8) |
| **Age in years** | | | | |
| ≤5 | 48 (15.9) | 54 (17.7) | 51 (16.8) | 153 (16.8) |
| 6-15 | 68 (22.5) | 69 (22.6) | 69 (22.7) | 206 (22.6) |
| >15 | 186 (61.6) | 183 (59.8) | 184 (60.5) | 553 (60.6) |
| **Education level** | | | | |
| No education | 93 (30.8) | 138 (45.1) | 101 (33.2) | 332 (36.4) |
| Primary education | 63 (20.9) | 67 (21.9) | 71 (23.4) | 201 (22.0) |
| Secondary sch. or above | 146 (48.3) | 101 (33.0) | 132 (43.4) | 379 (41.6) |
| **Occupation** | | | | |
| No occupation | 66 (21.9) | 52 (17) | 52 (17.1) | 170 (18.6) |
| Farmer | 23 (7.6) | 3 (1) | 10 (3.3) | 36 (3.9) |
| Government Office | 9 (3) | 20 (6.5) | 12 (3.9) | 41 (4.5) |
| Housemaid | 46 (15.2) | 57 (18.6) | 51 (16.8) | 154 (16.9) |
| Merchant/small business | 24 (7.9) | 21 (6.9) | 38 (12.5) | 83 (9.1) |
| Migrant labourer | 5 (1.7) | 6 (2) | 3 (1) | 14 (1.5) |
| Student | 97 (32.1) | 107 (35) | 109 (35.9) | 313 (34.3) |
| Other | 32 (10.6) | 40 (13.1) | 29 (9.5) | 101 (11.1) |
| Travel (in last 1 month) | 52 (17.2) | 46 (15.0) | 78 (25.7) | 176 (19.3) |
| Use of ITNs the previous night | 14 (4.6) | 258 (84.3) | 171 (56.3) | 443 (48.6) |
| *Mention fever as symptom | 186 (73.8) | 179 (82.1) | 213 (80.1) | 578 (78.5) |
| *Know about malaria mode of transmission | 197 (78.2) | 218 (100.0) | 243 (91.4) | 658 (89.4) |

* The total number of participants that responded to these questions were 736 (252 from Adama, 218 from Awash Sebat Kilo and 266 from Metehara).

to sustained exposure across age groups. Adama exhibited consistently low seroprevalence for both *P. falciparum* and *P. vivax*, with positive responses primarily restricted to individuals over 50 years of age, consistent with historical rather than recent transmission (Fig 2c and 2f).

A similar age-dependent increase in seroprevalence was observed for *P. vivax* using combined long-lived response markers (PvAMA1 and PvDBP RII PvRBP2) in both Metehara and Awash Sebat Kilo. These patterns indicate sustained transmission of *P. vivax* in these areas, with no evidence of recent decline. In contrast, sustained low seroprevalence across all age groups in Adama suggests limited *P. vivax* transmission over time (see Fig 2d, 2e, and 2f).

## Risk factors for seropositivity

Identifying the factors associated with *P. falciparum* seropositivity is essential for tailoring effective malaria control interventions. Using short-lived response markers, our GEE analysis identified key determinants of malaria exposure risk. Educational attainment showed a strong protective effect: individuals with primary education had 44% lower odds of infection exposure compared to those with no formal education (odds ratio [OR] = 0.56, 95% CI: 0.32–0.99; $p = 0.044$), while those with secondary education or higher experienced a 46% reduction in odds (OR = 0.54, 95% CI: 0.30–0.97; $p = 0.041$).

**Table 3. Fever and malaria prevalence in selected towns, Adama, Awash Sebat Kilo and Metehara, Ethiopia, January to March 2020.**

| Characteristics | City/Town | | | |
| --- | --- | --- | --- | --- |
| | Adama n(%) | Awash Sebat Kilo n(%) | Metehara n(%) | Total n(%) |
| **RDT Tested** | **255** | **261** | **258** | **774** |
| Negative | 255 | 259 | 254 | 768 |
| *P. falciparum* | 0 | 2 (0.8) | 3 (1.2) | 5 (0.7) |
| *P. vivax* | 0 | 0 | 0 | 0 |
| Mixed infection | 0 | 0 | 1 (0.4) | 1 (0.1) |
| **Reported fever** | **302** | **306** | **304** | **912** |
| Yes | 23 (7.6) | 16 (5.2) | 52 (17.1) | 91 (10) |
| **Travel history** | **302** | **306** | **304** | **912** |
| Yes | 52 (17.2) | 46 (15.0) | 78 (25.7) | 176 (19.3) |
| **qPCR test** | **257** | **259** | **257** | **773** |
| Negative | 240 (93.4) | 252 (97.3) | 250 (97.3) | 742 (96) |
| *P. falciparum* | 4 (1.6) | 5 (1.9) | 4 (1.6) | 13 (1.7) |
| *P. vivax* | 13 (5.1) | 2 (0.8) | 3 (1.2) | 18 (2.3) |
| **Parasite density (mean, Range)** | | | | |
| *P. falciparum* | 779.9 (101.1–1826.6) | 209.9 (121–293.4) | 302.1 (62.1–639) | 413.7 (62.1–1826.6) |
| *P. vivax* | 13.7 (9.3–22.2) | 8.5 (8.5–8.6) | 9.1 (8.8–9.3) | 12.4 (8.5–22.2) |
| **Serology Test** | **248** | **263** | **256** | **767** |
| *Positive for combined *P. falciparum* long-lived response markers | 14 (5.7) | 148 (56.3) | 98 (38.3) | 260 (33.9) |
| *Positive for combined *P. falciparum* short-lived response markers | 7 (2.8) | 96 (36.5) | 46 (18) | 149 (19.4) |
| *Positive for combined *P. vivax* long-lived response markers | 12 (4.8) | 102 (38.8) | 66 (25.8) | 180 (23.5) |
| *Positive for combined *P. vivax* short-lived response markers | 7 (2.8) | 74 (28.1) | 58 (22.7) | 139 (18.1) |

*P. falciparum* long-lived markers include (PfAMA1, PfMSP1_19, and GLURP R2) and *P. falciparum* short-lived markers (Etramp5Ag1, HSP40.Ag1 and GEXP18). *P. vivax* long-lived markers include (Pv DBP RII and PvAMA1) and *P. vivax* short-lived marker (PvMSP119 and PvEBP). Positive for combined markers" refers to samples that tested seropositive for at least one of the respective long-lived or short-lived markers for a given species (*P. falciparum* or *P. vivax*).

Environmental factors also played a significant role. Living near potential breeding habitats (within 1 km radius), such as any streams and rivers, was associated with nearly a threefold increase in the odds of *P. falciparum* seropositivity (OR = 2.80, 95% CI: 1.07–7.36; *p* = 0.037). Interestingly, some preventive measures were paradoxically associated with an increased risk of exposure. Individuals who reported sleeping under insecticide-treated nets (ITNs) had three times higher odds of exposure (OR = 3.03, 95% CI: 1.87–4.90; p < 0.001), while those living in households covered by indoor residual spraying (IRS) had more than double the odds (OR = 2.42, 95% CI: 1.53–3.85; p < 0.001). These findings may indicate that ITNs and IRS are being deployed in areas with higher transmission risk or may reflect limitations in the effectiveness of these interventions (Fig 3).

Comparable patterns were observed for *P. vivax*. Education remained a protective factor (primary: OR = 0.51, 95% CI: 0.29–0.87; *p* = 0.015; secondary or higher: OR = 0.54, 95% CI: 0.31–0.91; *p* = 0.021), while ITN use was similarly associated with elevated odds of seropositivity (OR = 2.77, 95% CI: 1.80–4.29; p < 0.001). Age also emerged as a significant risk factor: individuals aged over 15 years had four times higher odds of infection exposure compared to children under six (OR = 4.00 95% CI: 2.11–7.59; p < 0.001), likely reflecting increased outdoor and night-time activities that heighten exposure risk (Fig 3).

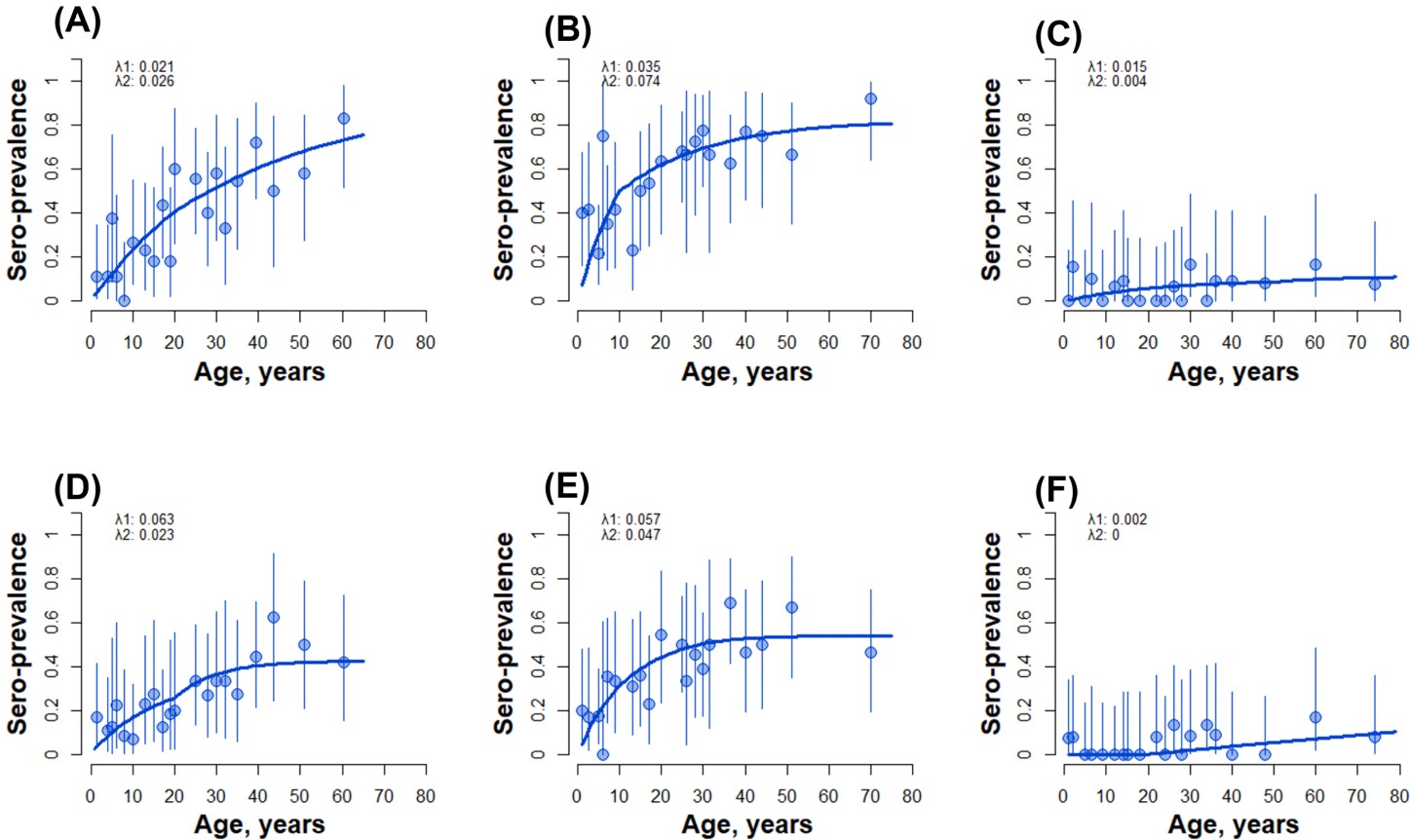

**Fig 2. Seroconversion rates (SCRs) for combined long-lived response markers for *P. falciparum* (A-C) (A: Metehara B: Awash Sebat Kilo C: Adama) and *P. vivax* (D-F) (D: Metehara E: Awash Sebat Kilo F: Adama).**

## Spatial clustering of exposure

Fig 4 depicts the distribution of *P. falciparum* and *P. vivax* seropositive individuals for recent marker across households, with Awash Sebat Kilo exhibiting the highest levels of exposure. Households in Awash Sebat Kilo had clusters of up to 7–8 seropositive individuals for both species, while smaller clusters (typically 2–4 individuals) were observed in Metehara. In contrast, Adama presented fewer clusters, mostly comprising 1–3 positive individuals per household.

Spatial autocorrelation analysis demonstrated significant clustering for both *P. falciparum* (Global Moran's I = 0.172650, z = 5.508310, $p < 0.0001$; S2a Fig) and *P. vivax* (Global Moran's I = 0.059049, z = 2.034681, p = 0.041883; S2b Fig). These indicate the presence of non-random spatial distribution patterns at the household level across the three study towns.

Further exploration using Anselin Local Moran's I (Cluster and Outlier Analysis) revealed distinct local spatial patterns (Fig 5) revealed distinct spatial clustering patterns. High seroprevalence areas (HIGH-HIGH clusters) were concentrated in Awash Sebat Kilo, while low-prevalence areas (LOW-LOW clusters) were mainly observed in Adama and Metehara. Additionally, spatial outliers (HIGH-LOW and LOW-HIGH clusters) were detected, representing localized anomalies in malaria seroprevalence patterns.

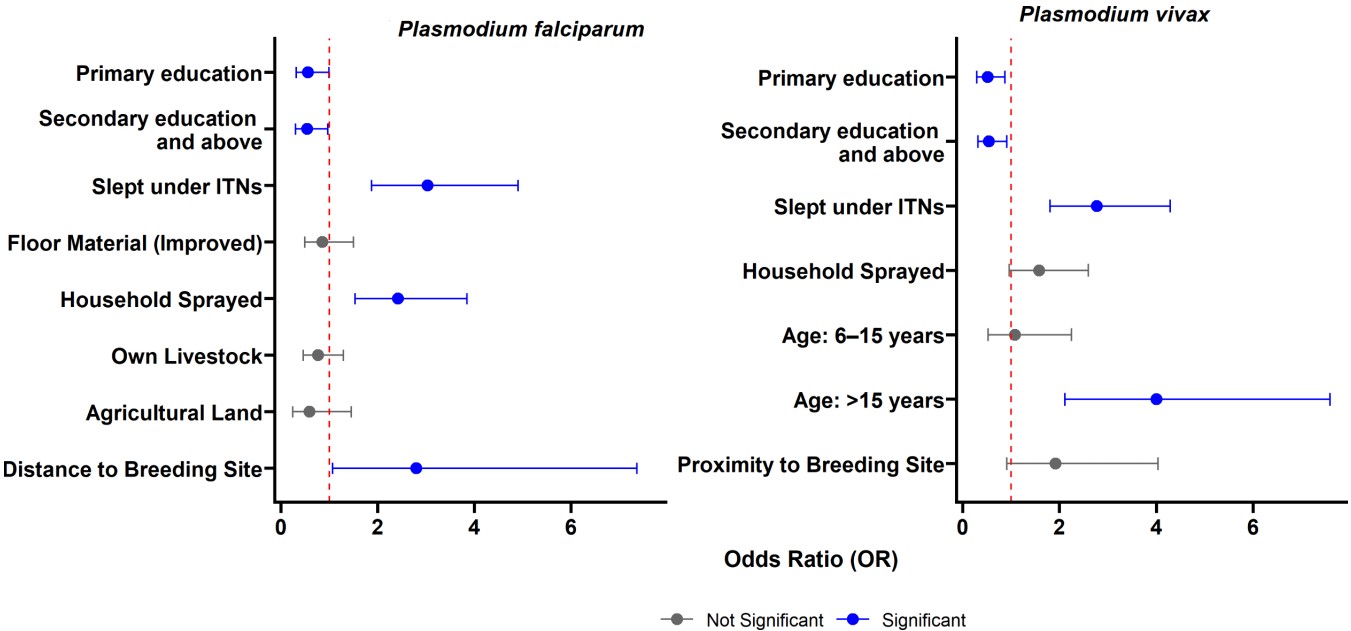

**Fig 3. Results of the final GEE model for risk factors associated with recent exposure to *P. falciparum and P. vivax*.** Factors with an odds ratio >1 are classified as risk factors and factors with an odds ratio <1 are protective factors.

## Discussion

This study provides insights into malaria transmission dynamics in three towns in Ethiopia: Adama, Metehara, and Awash Sebat Kilo, by combining molecular, serological, and spatial data. Despite very low infection rates (0.78% by RDT and 4% by PCR), we found evidence of persistent exposure, spatial clustering, and historical heterogeneity in transmission.

The discrepancy of RDT prevalence 0.78% (n = 6) versus qPCR prevalence 4% (n = 31) is due to the differences in diagnostic sensitivity. The result indicates the presence of low-density infections, that RDTs cannot detect (i.e., below the 100 parasite per microliter threshold versus 1–5 parasite per for PCR) [53]. This hidden reservoir has direct consequences as it can sustain residual transmission, mask hotspots, and cause over-optimistic assessments of elimination programme progress if active case screening of cases at elimination districts is guided by RDT results [54].

The malaria prevalence (with RDT 0.78%) observed in this study is consistent with RDT prevalence of 1.1% noted in urban settings in a nationally representative survey [14]. The lower prevalence observed in our study may be attributed to seasonal variations, as this study was conducted during the dry season, as well as temporal changes over time [55]. The marginally higher PCR prevalence of *P. vivax* (2.3%) than *P. falciparum* (1.7%) is typical in low transmission and dry seasons and in settings where *P. falciparum* has declined [56,57]. The biological resilience of *P. vivax,* notably its ability to relapse form dormant liver-stage hypnozoites [58], allows it to persist across seasons, contributing to asymptomatic and hidden reservoirs of infection [59].

Despite the low parasite prevalence, serological data provided an understanding of exposure and spatial heterogeneity. Overall, *P. falciparum* seroprevalence for recent exposure was higher than *P. vivax*, suggesting sustained transmission over time and is congruent with the more recent routine surveillance data for the study sites (see S1 Fig). The serology findings on prevalence of recent exposure to malaria aligned with low transmission levels for Adama. However, it indicated high transmission in Awash Sebat Kilo and moderate transmission in Metehara, which differs from the surveillance data. As serological markers are more stable for detecting malaria exposure and transmission level than surveillance data,

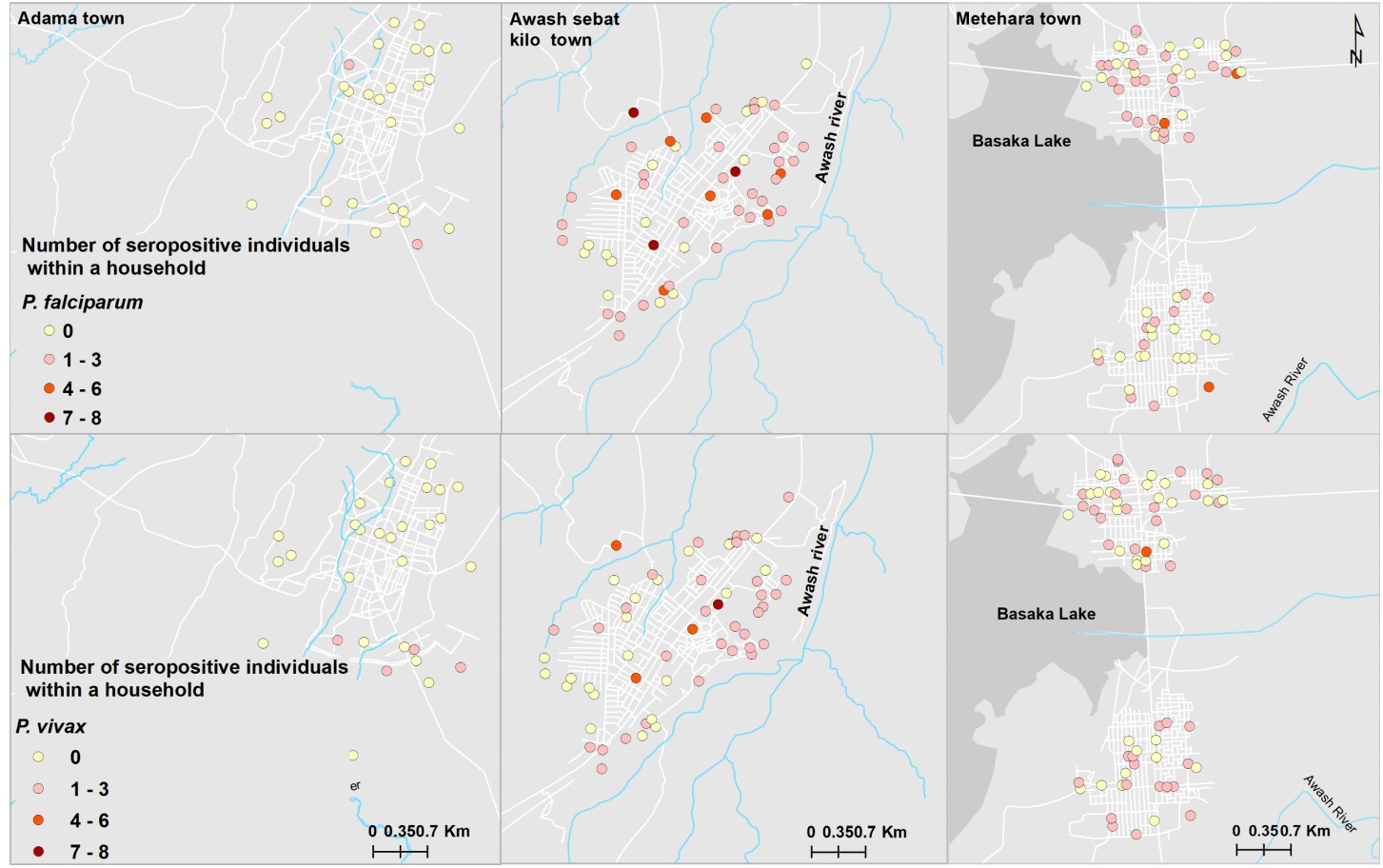

**Fig 4. Spatial distribution of visited households with recent exposure to *P. falciparum* and *P. vivax* in Adama, Metehara and Awash Sebat Kilo Towns [52].**

the discrepancy may be attributed to data quality, particularly the completeness of data in Afar compared to the Oromia Region, where these towns are situated [60].

Overall seroprevalence for long-lived markers for *P. falciparum* and *P. vivax* in Metehara and Awash Sebat Kilo were comparable to those reported in the national seroprevalence study [61]. The strong age-dependent increase in *P. falciparum* seropositivity for long term exposure in Metehara and Awash Sebat Kilo suggests cumulative exposure over time. The sharper decline in exposure among younger individuals in Metehara reflects more recent control success, whereas the lack of a similar trend in Awash Sebat Kilo points to persistent transmission. Adama exhibited low seropositivity across all age groups. This finding is consistent with previously observed and sustained reductions in malaria transmission and may also be attributed to the city's higher altitude (1,712 m) where transmission is usually is lower [61,62]. Based on the seroconversion rates curves, reduction in transmission could have occurred in Metehara around 20 years ago and 50 years ago in Adama, which corresponds with the history of control interventions in the country. Ethiopia has long and complex history of combating malaria, its 1st national malaria eradication program was launched in the 1950s, and among the dedicated malaria centers, Adama (formerly Nazreth) was established to support these efforts. Later in the beginning of the millennium, Ethiopia deployed 30,000 of health extension workers, introduced RDTs, LLINs, and artemisinin-based combination therapy [56]. The earlier changes in Adama may also correspond with rapid urbanization and improved infrastructure, which makes it one of the cities in the country.

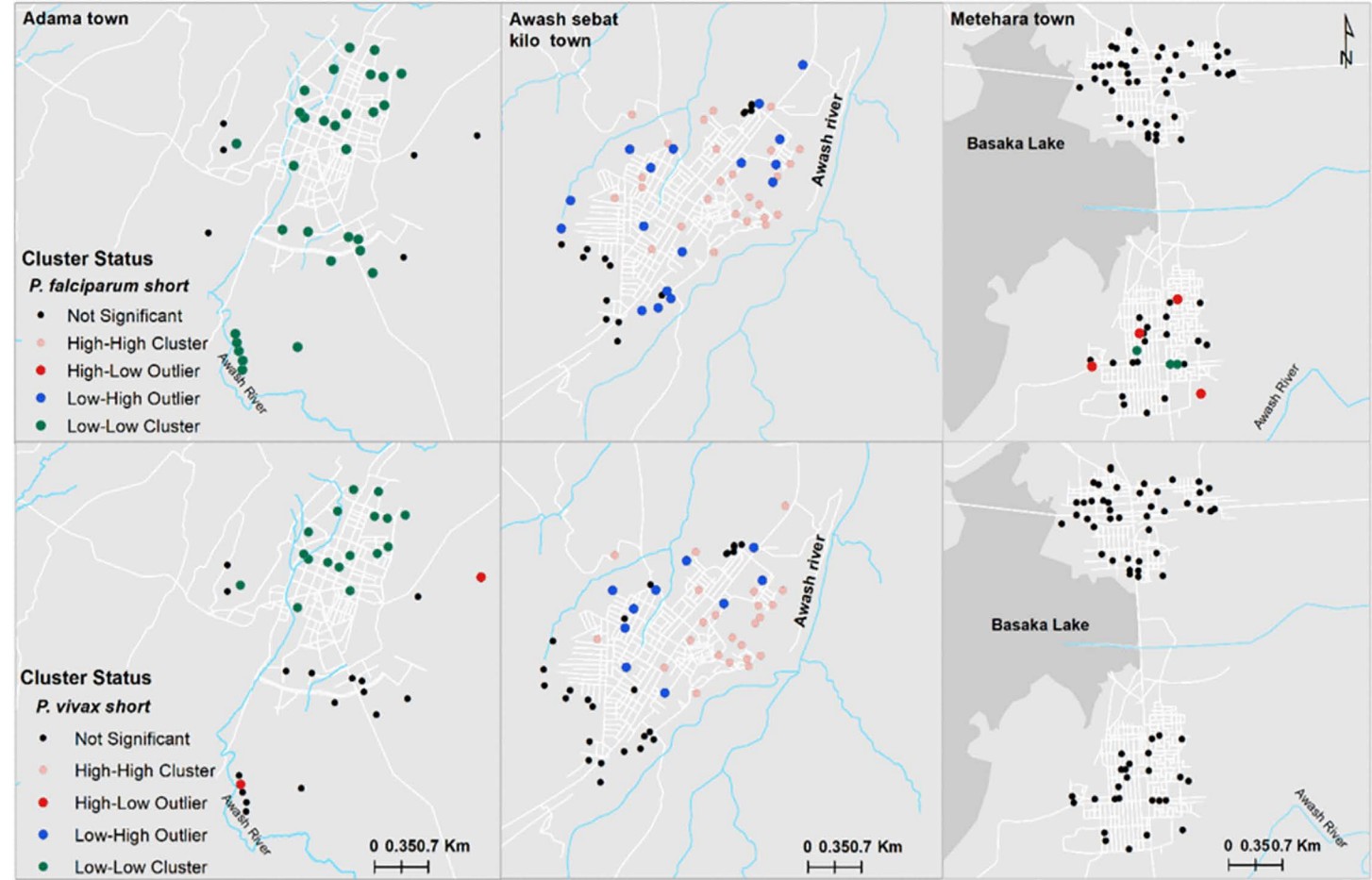

**Fig 5. Hotspot analysis of _P. falciparum_ and _P. vivax_ at the household level in Adama, Metehara, and Awash Sebat Kilo Towns [52].**

Spatial clustering of seropositive individuals for recent exposure further revealed micro-epidemiological variations. Significant hotspots were particularly detected for _P. falciparum_, and to a lesser extent, for _P. vivax_. This is in line with other studies in Adama [63]. Awash Sebat Kilo exhibited significant malaria hotspots, indicating high transmission intensity. In contrast, cold spots were more prominent in Adama and Metehara, where malaria prevalence was significantly lower. These patterns may reflect localized ecological risk factors such as proximity to vector breeding sites, differences in intervention coverage, or population mobility patterns affecting disease spread [5,56,64]. For instance, households located near rivers and streams were nearly three times more likely to be seropositive for _P. falciparum_, reinforcing the importance of targeted larval source management and environmental interventions.

Socioeconomic and behavioural factors were also important predictors of exposure to malaria [65–67]. Higher educational attainment was associated with significantly lower odds of malaria exposure for both _P. falciparum_ and _P. vivax_, underscoring the role of education in promoting health-seeking behaviour and adherence to preventive practices. This aligns with findings from other Ethiopian contexts, where education has been associated with better health literacy and increased adoption of malaria prevention measures [68,69]. Similarly, the increased odds of infection exposure among adults over 15 years may reflect occupational and behavioural risk that is common in Ethiopia [70,71], results in spending more time outdoors during peak mosquito activity.

One of the unexpected findings was the positive association between the use of preventive measures (ITNs and IRS) and higher odds of malaria exposure. These apparently contradictory results may be attributed to a combination of behavioural, intervention implementation and biological factors within the study areas. First, human behaviour such as inconsistent or improper ITN use could expose individuals to malaria infection [71]. Only about 50% of respondents reported sleeping under ITN the previous night, suggesting underutilization. Seasonal variations perceived low risk during the dry season, or discomfort in hot climates may discourage regular net use, leaving individuals vulnerable [72]. Secondly, the widespread resistance to deltamethrin among local *An. arabiansis* vector populations reported from several areas since 2008 [73,74] likely played a critical role in reducing the effectiveness of ITNs [75]. Studies have documented widespread resistance, enabling mosquitoes to survive contact with treated surfaces and continue transmitting malaria [74,76,77]. Thirdly, the quality and consistency of IRS campaigns can vary. Operational challenges such as incomplete spraying, poor timing, or use of less effective insecticides may lead to inadequate protection. Low coverage of IRS in these towns may contribute to increased exposure to malaria infection [78]. Additionally, the structural characteristics of houses, such as wall materials, can also influence the entry of mosquitoes in the houses [79]. Another plausible explanation is reverse causation: households or areas at higher risk of malaria may be prioritized for ITN distribution or IRS campaigns, creating an association where high-risk settings have greater intervention coverage. In such cases, the observed association does not imply that the interventions cause higher exposure but rather reflects targeted implementation in known hotspots. These findings echo similar results from a school-based survey in the Oromia Region [80] and other sub-Saharan African settings [81]. To address these challenges, malaria control programs must adopt more adaptive, data-driven strategies. This includes deploying next-generation tools such as dual-active ingredient nets, rotating insecticides for IRS, and enhancing community education to promote consistent ITN use.

A key limitation of this study is the relatively small sample size, which may have reduced the statistical power to detect associations between certain risk factors and malaria seropositivity, particularly when stratified by species or location. This limitation may also have contributed to wide confidence intervals for some variables and sero conversion rate, potentially obscuring more nuanced relationships. Additionally, the cross-sectional design restricts causal inferences and does not capture temporal or seasonal variations in malaria transmission. Furthermore, reliance on self-reported data for ITN use and IRS coverage introduces potential recall and reporting bias. Additionally, the three towns are in close proximity that may limit the generalizability of our findings to other locations. Future research should consider geographically representative, larger, longitudinal, or case-control studies incorporating environmental, entomological, and behavioural data to better identify household-level risk factors and optimize interventions in urban and peri-urban settings.

Finally, this study highlights the low malaria prevalence and the heterogeneous nature of malaria transmission in these three towns in Ethiopia. In addition, it demonstrates the value of integrating serological tests, specifically using short-lived marker and spatial data to uncover hidden reservoirs of malaria transmission, especially in low endemic and urban settings. As Ethiopia moves toward its malaria elimination goals, such integrative surveillance approaches are critical for identifying residual foci, refining interventions, and sustaining progress. Tailored strategies that consider local epidemiological, environmental, and behavioural contexts will be essential to achieving and maintaining malaria elimination.

## Supporting information

**S1 Fig. Malaria incidence trend in Adama, Metehara, and Awash Sebat Kilo towns.**
(TIF)

**S2 Fig. Spatial correlation analysis of *P. falciparum* (a) and *P. vivax* (b) in Adama, Metehara and Awash Sebat kilo towns.**
(TIF)

**S1 Table. List of coupled malaria antigens.**
(DOCX)

**S2 Table. Seroprevalence of malaria by species, antibody, and town (with 95%CI) for individuals less than 15 years of age.**
(DOCX)

**S1 File. Definition of variables.**
(DOCX)

## Acknowledgments

We extend our heartfelt gratitude to the Oromia and Afar Regional Health Bureaus, the East Shoa Zonal Health Office, and the Town Administrations of Adama, Metehara, and Awash Sebat Kilo for their invaluable support and collaboration throughout the course of this study. We are especially thankful to the community members and survey participants, whose generous participation made this research possible. We also wish to acknowledge Dr. Alemayehu Midexsa, Belendia Serda, and Semira Abdulmenan for their technical contributions to the development of data collection tools and data cleaning. Our sincere appreciation goes to Hailu Abera, Mikyas Gebremichael, and Senya Asfir for their diligent assistance with data collection in the field. Finally, we express special thanks to Dr. Jimee Hwang, Dr. Eric J. Tongren, and Dr. Mame Niang for their valuable scholarly input and feedback, which significantly enhanced the quality and rigor of our manuscript.

## Author contributions

**Conceptualization:** Hiwot Teka, Hassen Mamo, Fitsum Girma Tadesse.

**Data curation:** Hiwot Teka, Saron Fekadu, Legesse Alamerie Ejigu.

**Formal analysis:** Hiwot Teka, Saron Fekadu, Legesse Alamerie Ejigu, Solomon Sisay, Mulugeta Demisse.

**Funding acquisition:** Endalamaw Gadisa, Fitsum Girma Tadesse.

**Investigation:** Hiwot Teka, Saron Fekadu, Melat Abdo.

**Methodology:** Hiwot Teka, Legesse Alamerie Ejigu, Melat Abdo, Chris Drakeley, Isabel Byrne.

**Project administration:** Hiwot Teka.

**Resources:** Hiwot Teka.

**Software:** Legesse Alamerie Ejigu.

**Supervision:** Hassen Mamo, Chris Drakeley, Isabel Byrne, Girmay Medhin, Lemu Golassa, Endalamaw Gadisa, Fitsum Girma Tadesse.

**Visualization:** Legesse Alamerie Ejigu, Solomon Sisay, Mulugeta Demisse.

**Writing – original draft:** Hiwot Teka.

**Writing – review & editing:** Hiwot Teka, Saron Fekadu, Legesse Alamerie Ejigu, Solomon Sisay, Melat Abdo, Mulugeta Demisse, Hassen Mamo, Chris Drakeley, Isabel Byrne, Girmay Medhin, Lemu Golassa, Endalamaw Gadisa, Fitsum Girma Tadesse.

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
