## [Decision Letter · Decision Letter 0]

16 Oct 2025

Dear Dr. Namaga,

Thank you for submitting your manuscript to PLOS ONE. After careful consideration, we feel that it has merit but does not fully meet PLOS ONE’s publication criteria as it currently stands. Therefore, we invite you to submit a revised version of the manuscript that addresses the points raised during the review process.

We look forward to receiving your revised manuscript.

Kind regards,

Myat Htut Nyunt, MMedSc, PhD

Academic Editor

PLOS ONE

**Journal Requirements:**

1. When submitting your revision, we need you to address these additional requirements. Please ensure that your manuscript meets PLOS ONE's style requirements, including those for file naming. The PLOS ONE style templates can be found at https://journals.plos.org/plosone/s/file?id=wjVg/PLOSOne_formatting_sample_main_body.pdf and https://journals.plos.org/plosone/s/file?id=ba62/PLOSOne_formatting_sample_title_authors_affiliations.pdf 2. We note that the grant information you provided in the ‘Funding Information’ and ‘Financial Disclosure’ sections do not match.  When you resubmit, please ensure that you provide the correct grant numbers for the awards you received for your study in the ‘Funding Information’ section. 3. When completing the data availability statement of the submission form, you indicated that you will make your data available on acceptance. We strongly recommend all authors decide on a data sharing plan before acceptance, as the process can be lengthy and hold up publication timelines. Please note that, though access restrictions are acceptable now, your entire data will need to be made freely accessible if your manuscript is accepted for publication. This policy applies to all data except where public deposition would breach compliance with the protocol approved by your research ethics board. If you are unable to adhere to our open data policy, please kindly revise your statement to explain your reasoning and we will seek the editor's input on an exemption. Please be assured that, once you have provided your new statement, the assessment of your exemption will not hold up the peer review process. 4. Please amend either the title on the online submission form (via Edit Submission) or the title in the manuscript so that they are identical. 5. We note that Figures 1, 4 and 5 in your submission contain map images which may be copyrighted. All PLOS content is published under the Creative Commons Attribution License (CC BY 4.0), which means that the manuscript, images, and Supporting Information files will be freely available online, and any third party is permitted to access, download, copy, distribute, and use these materials in any way, even commercially, with proper attribution. For these reasons, we cannot publish previously copyrighted maps or satellite images created using proprietary data, such as Google software (Google Maps, Street View, and Earth). For more information, see our copyright guidelines: http://journals.plos.org/plosone/s/licenses-and-copyright. We require you to either present written permission from the copyright holder to publish these figures specifically under the CC BY 4.0 license, or remove the figures from your submission: a. You may seek permission from the original copyright holder of Figures 1, 4 and 5 to publish the content specifically under the CC BY 4.0 license.   We recommend that you contact the original copyright holder with the Content Permission Form (http://journals.plos.org/plosone/s/file?id=7c09/content-permission-form.pdf) and the following text:“I request permission for the open-access journal PLOS ONE to publish XXX under the Creative Commons Attribution License (CCAL) CC BY 4.0 (http://creativecommons.org/licenses/by/4.0/). Please be aware that this license allows unrestricted use and distribution, even commercially, by third parties. Please reply and provide explicit written permission to publish XXX under a CC BY license and complete the attached form.” Please upload the completed Content Permission Form or other proof of granted permissions as an "Other" file with your submission. In the figure caption of the copyrighted figure, please include the following text: “Reprinted from [ref] under a CC BY license, with permission from [name of publisher], original copyright [original copyright year].” b. If you are unable to obtain permission from the original copyright holder to publish these figures under the CC BY 4.0 license or if the copyright holder’s requirements are incompatible with the CC BY 4.0 license, please either i) remove the figure or ii) supply a replacement figure that complies with the CC BY 4.0 license. Please check copyright information on all replacement figures and update the figure caption with source information. If applicable, please specify in the figure caption text when a figure is similar but not identical to the original image and is therefore for illustrative purposes only.The following resources for replacing copyrighted map figures may be helpful: USGS National Map Viewer (public domain): http://viewer.nationalmap.gov/viewer/The Gateway to Astronaut Photography of Earth (public domain): http://eol.jsc.nasa.gov/sseop/clickmap/Maps at the CIA (public domain): https://www.cia.gov/library/publications/the-world-factbook/index.html and https://www.cia.gov/library/publications/cia-maps-publications/index.htmlNASA Earth Observatory (public domain): http://earthobservatory.nasa.gov/Landsat:
http://landsat.visibleearth.nasa.gov/USGS EROS (Earth Resources Observatory and Science (EROS) Center) (public domain): http://eros.usgs.gov/#Natural Earth (public domain): http://www.naturalearthdata.com/ 6. We notice that your supplementary figures are uploaded with the file type 'Figure'. Please amend the file type to 'Supporting Information'. Please ensure that each Supporting Information file has a legend listed in the manuscript after the references list. 7. Please include captions for your Supporting Information files at the end of your manuscript, and update any in-text citations to match accordingly. Please see our Supporting Information guidelines for more information: http://journals.plos.org/plosone/s/supporting-information. 8. If the reviewer comments include a recommendation to cite specific previously published works, please review and evaluate these publications to determine whether they are relevant and should be cited. There is no requirement to cite these works unless the editor has indicated otherwise. 

Reviewers' comments:

**Comments to the Author**

1. Is the manuscript technically sound, and do the data support the conclusions?

Reviewer #1: Yes

Reviewer #2: Yes

Reviewer #3: Partly

2. Has the statistical analysis been performed appropriately and rigorously?

Reviewer #1: Yes

Reviewer #2: N/A

Reviewer #3: Yes

3. Have the authors made all data underlying the findings in their manuscript fully available?

Reviewer #1: Yes

Reviewer #2: Yes

Reviewer #3: Yes

4. Is the manuscript presented in an intelligible fashion and written in standard English?

Reviewer #1: Yes

Reviewer #2: Yes

Reviewer #3: Yes

**Reviewer #1: ** The authors have produced an interesting, a highly exhaustive, timely, and impactful study on emerging challenges related to urban malaria in Ethiopia. This work has the potential to significantly contribute to the ongoing malaria elimination efforts in the country. Overall assessment: The manuscript is requiring only minor revisions. The authors have presented a sound study that requires only minor adjustments to clarity, formatting, and the discussion of key findings. And Answer to some questions.

**Reviewer #2:**  Date: October 14, 2025

MS title: Heterogeneity of Malaria in Urban Settings in Ethiopia: A Sero-prevalence and Risk

Factor Study

General:

I have carefully read the document it is well organized and written. The authors investigated urban malaria using Microscopy, PCR and serology methods and reported heterogeneity and recommended tailored intervention approaches.

While the work is good its main drawback is the sampling sites, the three towns selected are relatively in close spatial proximity with similar eco-epidemiology characteristics and all located in the single corridor that connect Djibouti and Addis. They are also connected and supported by single river. Not sure if this is purposive, otherwise, this limits the generalizability of the study.

Comments: Considering the below comments may improve the MS

- Authors indicate Ethiopia is the 4th largest contributor globally and later in line 67 they state elimination goal by 2030, is it feasible and relevant mentioning it in the MS? And would this elimination plan for 2030 a targeted elimination in low transmission settings?

- Authors may want to indicate intervention interruption among the factors for malaria dynamics ex. Current outbreak/resurgence may have been influenced by intervention interruption

- Line 50-52 sounds to have some redundancy with first paragraph

- The data collection period jan to march is out of transmission season, unless there is a reason this may bias the burden estimation

- The study uses pfhrp2 RDT but does not consider the effect of hrp discussion that may affect diagnosis accuracy

- Consider uniformity in methods either describe details or reference ex line 134 either write the extraction procedure in short or make it full. Similar comment may hold for PCR and serology assays

- Line 136-139 consider revising language

- Line 156 antigen coupling, this is relevant either describe it or reference it

- Line 172 define MFI

- Line 207 what does approved by AHRI means consider language revision

- Line 223 television add % for consistency

- Line 225/26 consider the use of only, better reporting as is

- Table 1 the word improved is confusing ex. What is improved roof …is it corrugated iron roof or what?

- Line 233 consider median age and IQR

- Line 304/6 would the relation of infection vs IRS/ITN may be affected by previous exposure or infection history

- Line 341 discussion consider referencing this document : 10.1186/s12936-019-2874-z.

- Line 377 is 1712 that high to affect malaria transmission ?

- Line 395what about increase on age as a risk factor for infection?

- Line 405 resistance to deltamethrin may reduce efficacy but cannot increase infection, please elaborate

- Try to increase picture resolution for all and put pictures to scale

- Consider typo totmal/Total supplement figure 1

**Reviewer #3:**  Comments

1) This is an interesting paper exploring the use of serologic markers for fine scale epidemiological mapping of malaria transmission heterogeneity in urban settings. It also aimed to identify risk factors.

Perhaps the title should state “Heterogeneity of malaria transmission….” adding the key word “transmission”. It is also possible to suggest a modified title since heterogeneity is natural and expected. Just an example “A sero-prevalence and risk factor study of urban malaria in Ethiopia”.

2) While exploring potential risk factors, the authors could have taken account of physico-environmental (altitude, mean monthly temperature, etc.), population size/density, and vector parameters and species in the analysis of the data. It seems all the three towns have similar agro-climatic setting and are close to Awash River and the associated irrigation agriculture schemes. Just wondering if the choice of study sites was ideal for the intended purpose. High transmission towns or a high transmission season could have been opted.

3) The authors have rightly pointed out the key limitation of the study, i.e., sample size. First and foremost, the geographic variations (location) are not accounted for, and the assumption about prevalence (10% PCR positivity) and household size of 3 persons per HH was not well thought of. It is also unclear how household numbers were partitioned between the three towns, with higher number of households sampled in Awash where the total population was the least and ultimately a higher prevalence was found in this town. These estimates would probably be valid for the high transmission season in these localities. In addition, there was a high dropout rate (81 households not sampled). It seems the risk of bias could be high. Given this, the authors could have softened the tone of the conclusions. It is impossible to make hard conclusions and at the same time highlighting significant limitations. In line 342, they mention that the study has given ‘new insight about malaria transmission dynamics”. Furthermore, in lines 354/355, they also mention the serological data has allowed a deeper understanding of exposure. I think that study has given an insight that might help future large-scale studies, but it is far from the provision of a deep understanding of exposure.

4) Some discussion points are speculative and unsharpened. For instance, the issue of data quality (lines 362-363) implicates poor quality in Afar and better quality in Oromia. While this assessment could be true, it barely qualifies as a valid statement when not supported by evidence. The authors are seeking potential explanations for observed events, but for simplest explanations to complex phenomenon. In line 366, they refer to the success of control efforts in Metehara to explain recent declines (among young individuals) in malaria prevalence. This also requires a proper assessment of control efforts and its impact, but also by extending the assessment of seroconversion rates over different time points especially as we know malaria transmission varies seasonally and geographically. The success of a control program can be better evidenced preferably by measuring and finding a decline in sero-conversion rates over repeated time points, and statistically by determining the likelihood ratio. Similarly in lines 413-417, the discussion of reverse causation in explaining the higher odds of exposure to malaria in protected individuals/communities is not successful. It might just help posing questions whether such situations are explanatory.

5) In lines 349-351, the authors also discuss the higher prevalence of P. vivax than P. falciparum as a typical phenomenon in low transmission seasons. This is probably partially true. But, since most of the tested individuals were asymptomatic, there is also high chance that asymptomatic parasitemia is contributed largely by P. vivax even in high transmission seasons or localities.

6) There were 91 individuals (10%) who had fever in the two weeks period prior to sampling. It would have helped what proportion of these were RDT/Serology/PCR positive for malaria cf. total population. A similar analysis of data for individuals who had slept under the net cf. those who have not, might also give an insight of exposure patterns. Such data seems to be available in figure 3, but the title and the legends of figure 3 do not indicate what tests were used to determine exposure and to ultimately calculate OR. Such data might help to resolve the discussion of paradoxes, e.g., the higher odds of exposure in protected individuals and provide opportunity for alternative explanations. It is also possible that sero-positivity may be associated with protection from repeated exposure (due to acquired immunity), and the increased odds may be obscure rather than paradoxical.

Minor comments

1) In line 53, the number of malaria cases is given in 6-digit number and then the word “million” is included. In addition, the authors sequence the year and number of cases (i.e., “In 2023, 5,677,912”). This could be tidied by restructuring the sentence and alternating the sequence of numbers. For example, it can be stated as “In the Ethiopian HMIS, there were reports of 5,677,912 malaria cases in 2023, accounting for ……….,”. Similarly, in line 222-223, the statement “Most households owned mobile phones 95% (n=186) and a television (n=164)” which sequences words and percentage. I suggest the sentence to read as “Most households owned mobile phones (n=186, 94.9%) and television (n=164, 83.7%). This is a common problem in the many manuscripts that we have reviewed previously. Another example is in lines 226-227, whereby sequencing is noted. The statement “Most households had improved roofing 87% (n=171) and flooring 76% (n=149) materials, 31% (n=61). The problem is the sequence “flooring 76%” and “roofing 87%”, and the 31% is supposed to be for eaves not included in the statement. It is better written as “Most households had improved roofing (n=171, 87%) and flooring (n=149, 76%) materials and eaves (n=61, 31.1%).

2) In lines 267-269, seroconversion rates of 0.15, 0.035, 0.002, 0.063 are mentioned. Are these percentages or proportions? They appear to be “proportions” without the % symbol.

Thank you!

**Do you want your identity to be public for this peer review?** For information about this choice, including consent withdrawal, please see our Privacy Policy

Reviewer #1: No

Reviewer #2: No

Reviewer #3: No

---

## [Author Response · Author response to Decision Letter 1]

5 Dec 2025

Response to the Editor

We have edited the following in line with the guidelines

1. Edited the title of the manuscript in sentence case

2. Edited the affiliations of the authors as per the guideline

3. Included the supporting information section after the reference

4. Edited the title of the supporting information

5. supporting information are uploaded separately

We have included the below statement for funding information. There is no funding/Award number as this is a core budget. Let us know where to make the corrections.

Norwegian Agency for Development Cooperation (NORAD) and Sweden International Development Agency (Sida) provided core funding for Armauer Hansen Research Institute. The funders had no role in study design, data collection and analysis, decision to publish, or preparation of this manuscript.

All authors agreed to make the data to be freely accessible. All data will be made available on Dryad (linked with the ORCID: https://orcid.org/0000-0003-1931-1442). The R codes used to run the analyses reported in this study will be made available at https://github.com/legessealamerie/Urban_malaria_Ethiopia.

We have edited the title and ensured that it is identical with the manuscript.

5. We note that Figures 1, 4 and 5 in your submission contain map images which may be copyrighted. All PLOS content is published under the Creative Commons Attribution License (CC BY 4.0), which means that the manuscript, images, and Supporting Information files will be freely available online, and any third party is permitted to access, download, copy, distribute, and use these materials in any way, even commercially, with proper attribution. For these reasons, we cannot publish previously copyrighted maps or satellite images created using proprietary data, such as Google software (Google Maps, Street View, and Earth). For more information, see our copyright guidelines: http://journals.plos.org/plosone/s/licenses-and-copyright.

We require you to either present written permission from the copyright holder to publish these figures specifically under the CC BY 4.0 license, or remove the figures from your submission:

a. You may seek permission from the original copyright holder of Figures 1, 4 and 5 to publish the content specifically under the CC BY 4.0 license.

Thank you for the suggestions. We confirm that all maps presented in Figures 1, 4, and 5 were entirely generated by the authors in ArcGIS /ArcMap, using open-access basemaps and publicly available data sources that are fully compatible with the CC BY 4.0 license. The basemaps were added directly through the ArcGIS “Add Basemap” feature, specifically the Esri World Light Gray Canvas Base and OpenStreetMap Basemap services, which incorporate data from Esri, HERE, Garmin, FAO, NOAA, USGS, and OpenStreetMap contributors. No Google or other proprietary imagery was used at any stage. Full attributions for these basemap sources are included in the figure legends and reference list, ensuring full compliance with PLOS’s open-license requirements.

6. We notice that your supplementary figures are uploaded with the file type 'Figure'. Please amend the file type to 'Supporting Information'. Please ensure that each Supporting Information file has a legend listed in the manuscript after the references list.

We have included supporting Information files after the reference and included the legends.

We have included supporting Information files after the reference, and update any in-text citations to match accordingly.

We accepted the recommendation and ensured suggested citations are appropriately included.

---

## [Decision Letter · Decision Letter 1]

18 Jan 2026

Heterogeneity of malaria transmission in urban settings in Ethiopia: A seroprevalence and risk factor analysis.

PONE-D-25-32925R1

Dear Dr. Namaga,

We’re pleased to inform you that your manuscript has been judged scientifically suitable for publication and will be formally accepted for publication once it meets all outstanding technical requirements.

Kind regards,

Myat Htut Nyunt, MMedSc, PhD

Academic Editor

PLOS One

Additional Editor Comments (optional):

Reviewers' comments:

Reviewer's Responses to Questions

**Comments to the Author**

Reviewer #1: All comments have been addressed

Reviewer #2: All comments have been addressed

2. Is the manuscript technically sound, and do the data support the conclusions?

Reviewer #1: Yes

Reviewer #2: Yes

3. Has the statistical analysis been performed appropriately and rigorously?

Reviewer #1: Yes

Reviewer #2: (No Response)

4. Have the authors made all data underlying the findings in their manuscript fully available?

Reviewer #1: Yes

Reviewer #2: (No Response)

5. Is the manuscript presented in an intelligible fashion and written in standard English?

Reviewer #1: Yes

Reviewer #2: (No Response)

Reviewer #1: MY FEEDBACK FOR PLOS ONE:MS reference number: PONE-D-25-32925R1.

Corresponding Author: Hiwot Teka Namaga, Addis Ababa University, Addis Ababa, ETHIOPIA

My general reflection on the Work:

I reiterate my previous feedback’s introduction as follows: -

the authors have produced a highly exhaustive, timely, and impactful study on the emerging challenges of urban malaria in Ethiopia. This work has the potential to significantly contribute to the country’s ongoing malaria elimination efforts.

The authors have also addressed my queries point-by-point, providing sound scientific and biological justifications. I appreciate the thoroughness of their literature review and the resulting improvements to the manuscript. Given the depth of my previous feedback, I find the authors' current revisions to be more than satisfactory.

Minor Suggestion

My only remaining suggestion concerns the notation for the Adjusted Odds Ratio (AOR). Currently, the authors use "OR," which typically denotes a Crude Odds Ratio (COR). As inferences should be based on the adjusted model, I recommend the authors consistently use the abbreviation AOR to clearly indicate that the measure accounts for potential confounders.

With this minor adjustment, I recommend the manuscript for publication.

Reviewer #2: (No Response)

**Do you want your identity to be public for this peer review?** For information about this choice, including consent withdrawal, please see our Privacy Policy

Reviewer #1: No

Reviewer #2: No

<gdiv id="ginger-floatingG-container" style="position: absolute; top: 0px; left: 0px; display: none;"><gdiv class="ginger-floatingG ginger-floatingG-closed ginger-floatingG-posdown ginger-floatingG-loading" style="display: block; left: 658.5px; top: 171px; z-index: 51;"><gdiv class="ginger-floatingG-disabled-main"><gdiv class="ginger-floatingG-bar-tool-tooltip ginger-floatingG-bar-tool-tooltip-enable">Enable Ginger</gdiv></gdiv><gdiv class="ginger-floatingG-offline-main"><gdiv class="ginger-floatingG-bar-tool-tooltip">*Cannot connect to Ginger* Check your internet connection

or reload the browser</gdiv></gdiv><gdiv class="ginger-floatingG-enabled-main"><gdiv class="ginger-floatingG-bar"><gdiv class="ginger-floatingG-bar-tool ginger-floatingG-bar-tool-disable"><ga></ga><gdiv class="ginger-floatingG-bar-tool-tooltip">Disable Ginger</gdiv></gdiv><gdiv class="ginger-floatingG-bar-tool ginger-floatingG-bar-tool-correction-tip ginger-floatingG-bar-tool-correction-tip_hidden"><ga>?</ga><gdiv class="ginger-floatingG-bar-tool-tooltip">How to use Ginger</gdiv></gdiv><gdiv class="ginger-floatingG-bar-tool ginger-floatingG-bar-tool-rephrase ginger-floatingG-bar-tool-rephrase_big-circle"><ga class="ginger-floatingG-bar-tool-rephrase__btn" id="ginger__floatingG-bar-tool-rephrase__btn">Rephrase</ga><gdiv class="ginger-floatingG-bar-tool-tooltip ginger-floatingG-bar-tool-tooltip_rephrase">Rephrase with Ginger (Ctrl+Alt+E)</gdiv></gdiv><gdiv class="ginger-floatingG-bar-tool ginger-floatingG-bar-tool-mistakes"><ga>0</ga><gdiv class="ginger-floatingG-bar-tool-tooltip">Log in to edit with Ginger</gdiv></gdiv></gdiv></gdiv><gdiv class="ginger-floatingG__loading-popup">Ginger is checking your text for mistakes...</gdiv><gdiv class="ginger-floatingG__disabling-popup " style="display: none;"><button class="ginger-floatingG__disabling-popup-button">Disable Ginger in this text field</button><button class="ginger-floatingG__disabling-popup-button">Disable Ginger on this website</button></gdiv><gdiv class="ginger-floatingG-contentPopup" style="display: none;"><gdiv class="ginger-floatingG-contentPopup-wrap "><ga class="ginger-floatingG-contentPopup-close ">×</ga><gdiv class="ginger-floatingG-contentPopup-frame"><iframe scrolling="no"></iframe></gdiv></gdiv></gdiv></gdiv></gdiv>

---

## [Editor Report · Acceptance letter]

PONE-D-25-32925R1

PLOS One

Dear Dr. Teka,

I'm pleased to inform you that your manuscript has been deemed suitable for publication in PLOS One. Congratulations! Your manuscript is now being handed over to our production team.

Kind regards,

on behalf of

Dr. Myat Htut Nyunt

Academic Editor

PLOS One